# Efficient Separation of the Methoxyfuranocoumarins Peucedanin, 8-Methoxypeucedanin, and Bergapten by Centrifugal Partition Chromatography (CPC)

**DOI:** 10.3390/molecules28041923

**Published:** 2023-02-17

**Authors:** Magdalena Bartnik

**Affiliations:** Department of Pharmacognosy with Medicinal Plants Garden, Medical University of Lublin, Chodźki 1 St., 20-093 Lublin, Poland; mbartnik65@gmail.com

**Keywords:** centrifugal partition chromatography (CPC), semi-preparative CPC, peucedanin, 8-methoxypeucedanin, bergapten, *Peucedanum tauricum*

## Abstract

Pure methoxyfuranocoumarins were isolated from a crude petroleum ether extract (CPE; Soxleth extraction efficiency 12.28%) from fruits of *Peucedanum tauricum* MB. (Apiaceae) by counter-current chromatography in a hydrostatic equilibrium system (centrifugal partition chromatography—CPC). The optimized biphasic solvent system composed of *n*-heptane-ethyl acetate-methanol-water (5:2:5:2; *v*/*v*/*v*/*v*) in the ascending mode of elution was used (3 mL/min, 1600 rpm). In the single run, peucedanin (P), 8-methoxypeucedanin (8MP), and bergapten (5MOP) were obtained as pure as 95.6%, 98.1%, and c.a. 100%, respectively. The carefully optimized and developed CPC was effectively transferred from the analytical to the semi-preparative scale (where 20 mg and 150 mg of CPE were loaded, respectively). Identification and quantitative analysis of methoxyfuranocoumarins was carried out in the plant material, in the CPE, and in individual CPC fractions by use of validated high-performance liquid chromatography with diode array detection and mass spectrometry (HPLC-DAD-ESI-MS). For the separation steps, the extraction/isolation recovery was calculated. In this case, CPC proved to be an effective tool for the simultaneous isolation and separation of P, 8MP, and 5MOP from a multicomponent plant matrix, without additional pre-purification steps. The high purity of the obtained plant metabolites makes it possible to consider their use in pharmacological or biological studies.

## 1. Introduction

In these current times, we are witnessing a renaissance of natural products as drug candidates [1,2]. Natural products and their semi-synthetic analogs have been considered as promising natural medicines, clinical therapeutics candidates, and also adjuvants in modern therapies. Due to the complexity of the natural plant matrix, it is still a challenge to separate and isolate pure compounds from these sources in an efficient, rapid, cost effective, and also environmentally favorable way.

Counter-current chromatography (CCC) [3,4,5] is a separation technology that combines the features of partition chromatography and liquid–liquid extraction [6] and therefore possesses a high adaptability to different separation tasks, especially for the separation of natural products [7,8,9]. This liquid–liquid isolation technique, without a solid sorbent in the system [10], ensures high efficiency and eliminates the irreversible adsorption of the sample on the solid stationary phase, which results in almost 100% of its recovery [6,11]. First developed by Ito et al. [12], it is still being expanded [13]. Due to its low solvent consumption, high load capacity, and easy scale-up, CCC provides an convenient tool to obtain pure compounds on the analytical and preparative scale of separations [11,14]. In CCC, two types of devices are commonly in use: high-speed counter current chromatographs (HSCCC) and centrifugal partition chromatographs (CPC); however, both utilize two phases (mostly binary phases) composed from immiscible liquids, one of which serves as a stationary phase, which is kept on the column by special forces, and the second, pumped through the column, works as the mobile phase [10,15]. In both types of apparatus, when the mobile phase is pumped through the column, the sample components are partitioned between the two phases, but the efficiency and the separation system are not exactly the same due to the different mechanisms of separation [7,15,16]. As has recently been reported, in some cases, the use of hydrostatic CCC enables efficient isolation and separation of plant secondary metabolites, as compared to the HSCCC process [17]. However, in each case, one of the most important tasks before CCC is still the selection of a biphasic liquid system and the mobile and stationary phases [16]. For each plant extract, which we intend to fractionate, and also for each of the separated group of compounds, these parameters are of great importance and should be carefully optimized. They should provide a suitable range of the partition coefficient values (K), of which the value for efficient separation of the target compounds should be close to 1 (preferably) or between 0.2–5.0. No less important is the separation factor between the two components (Kα = K1/K2, K1 > K2), which should be greater than 1.5—a smaller value, as in conventional chromatography, may result in the loss of peak resolution [15,18]. In practice, in CPC, only a restricted number of solvents is used, most frequently *n*-heptane or *n*-hexane, ethyl acetate, methanol or ethanol, and water [13], which enables separation of compounds with a wide range of polarity [19]. However, it is strongly recommended, in each case of a new plant matrix fractionation, to also optimize such parameters such as the flow rate, rotation/revolution speed, and the type of elution (ascending, descending mode, or, e.g., elution–extrusion), as they have a big influence on the final effect of successful isolation of pure components [7,13,16].

Plants from the *Peucedanum* genus are from ancient times and are a part of the system of natural medicine, e.g., in European and Asian countries, and in particular, they are used in traditional Chinese medicine (TCM), where plant parts or extracts from, e.g., *P. praeruptorum*, *P. japonicum*, *P. decursivum*, *P. officinale*, and *P. alsaticum* are used as natural drugs [20,21,22]. These and many other *Peucedanum* species have been phytochemically analyzed, with a focus on the isolation of active metabolites, mostly essential oil constituents and benzo-α-pyrone derivatives [20,23,24] and on their pharmacological activity [20,25,26,27,28,29]. The latter constituents of the *Peucedanum* plants, from which psoralene-type furanocoumarins are of great importance are comprehensively analyzed nowadays for new applications, especially as anticancer therapeutics, anxiolytic agents, chemotherapy adjuvants, or chemo-preventive agents [21,30,31,32,33,34,35]. *Peucedanum tauricum* MB. is an endemic perennial plant with a restricted occurrence in the Crimea, Caucasus, and Romania [36,37,38]. This species has been found to be an interesting source of secondary metabolites such as new guaiene-type sesquiterpenoids [39,40] and methoxyfuranocoumarins [39,41,42,43], such as 8-methoxypeucedanin (8MP), peucedanin (P), and bergapten (5MOP) (see the chemical formula in Figure 1). All of them have been previously isolated from this plant matrix, and although the ultimately developed isolation procedures made it possible to obtain compounds with high purity, in each case, however, they were time-consuming and involved multiple stages [39,40,41,42,44,45]. The estimation of the content of secondary metabolites (such as essential oil components, flavonoids, phenolic acids, and coumarins) in extracts of different parts of *P. tauricum* has been carried out in different ontogenetic stages (flowers, mature and immature fruits, and each time simultaneously in leaves). For this purpose, a validated SPE/HPLC-DAD procedure was developed and applied [39,43]. This evaluation confirmed that fruits (mature and also immature) are rich in methoxyfuranocoumarins mostly P, 8MP, and 5MOP [39,43].

It is worth noting that natural coumarins and their analogues are still being intensively researched (they are also the subject of many patent applications) due to their broad spectrum of activity allowing their use in antiviral, anticancer, antioxidant, and anti-inflammatory protocols as natural substances with relatively low toxicity and lower drug-resistance properties [46]. Some methoxycoumarins are important plant drugs, such as, e.g., xanthotoxin (8-methoxypsoralen), which has been used in psoriasis and vitiligo treatment for many years. Recently, the influence of this natural substance on apoptosis induction by intrinsic and extrinsic pathways and suppression of cell growth of SK-N-AS neuroblastoma and SW620 metastatic colon cancer cells has been discovered [33]. The rare furanocoumarins P and 8MP have recently gained some attention [45,47]; however, only a few reports concerning their biological activity have appeared to date. Peucedanin, studied previously, was found to be active against Ehrlich’s tumors and human hepatic carcinoma HepG2 cells [48]. In addition, the activity of this compound as a blocker of Ca^2+^ canals has been confirmed [25] and also activation of the early stages of apoptosis (in an Annexin V-Cy3 test) with no necrosis effect, the decrease of heat-shock proteins Hsp 27 and 72 levels, and morphological changes in the tested HeLa cells, as has previously been reported [28,39]. Some interesting interaction between P and biological membranes (lipid monolayers) has been preliminary studied [47], and an interesting structural organization of crystals of this furanocoumarin in polar solvents was found [45]. In recent studies, P and 8MP have been found to be moderate activators of apoptosis, blocking the G1 phase of the cell cycle and showing anti-proliferative activity in promyelocytic HeLa cells [29]. Anti-bacterial activity against *Staphylococcus*, *Pseudomonas*, and *Klebsiella* species was found for P and moderate activity in the case of 8MP was also detected [49]. Bergapten was found to be cytotoxic and it exerts anti-proliferative activity and a pro-apoptotic effect in human HepG2 cell lines by arresting the G2/M phase of the cell cycle [50]. In addition, a weak influence of 5MOP on apoptosis induction and heat-shock proteins expression in HeLa cells has previously been reported [28,29,39,49].

As it remains important to develop efficient methods for the extraction, isolation, and purification of furanocoumarins from plant matrices [51], among other chromatographic techniques, CCC plays an important role in this case as a still evolving and effective methodology [52,53,54]. The different types of coumarins (simple, furano-, and pyranocoumarins) have been previously successfully isolated by use of centrifugal partition chromatography (CPC) and high-speed counter-current chromatography (HSCCC) [26,27,54,55,56,57,58]. However, there has only been one report concerning the isolation of peucedanin by HSCCC found to date [49], and according to our knowledge, there are no reports about isolation of 8-methoxypeucedanin by use of CCC, without any additional techniques or purification steps. Interestingly, only a few reports are available up to this time about isolation of 5MOP by HSCCC and hyphenated couplings of CCC with preparative HPLC [59,60,61,62]. Up to this date, no reports about the use of CPC for isolation of 5MOP, P, and 8MP and also no reports about the isolation of these three coumarins by the CCC method simultaneously from one plant source are available. The goal of the presented work is the development of hydrostatic counter-current chromatography (CPC), optimisation of the process and the next application for isolation of pharmacologically important methoxyfuranocoumarins: peucedanin, 8-methoxypeucedanin, and bergapten from a complexed matrix of plant origin—*Peucedanum tauricum* fruit crude extract, in the single run, without any pre-purification and additional procedures. The transfer from the analytical to semi-preparative scale of isolation is the next step to the successful isolation of these coumarins from a complicated matrix. Extraction efficiency for each step of the process was monitored by HPLC and identitying isolated compounds was additionally confirmed by ESI-MS analysis.

## 2. Results

### 2.1. Extraction of Plant Material

As the result of Soxhlet extraction from 25 g of plant material, petroleum ether extract was collected and preliminary concentrated. After filtering (0.45 µm Cronus PTFE syringe filters; SMI-LabHut Ltd., Glouchester, UK) and evaporating the solvent, 3.07 g of crude concentrated petroleum ether extract from the fruits (CPE) was finally gained and subjected to the next steps of isolation. The efficiency of this step of isolation was calculated as 12.28%.

### 2.2. CPC Method Optimisation

The optimization of the two-phase system for targeted compounds in CPE was performed, and Arizona heptane systems I–VI (Table 1) containing *n*Hp:E:M:W (in ascending or in descending mode) were tested (with compilations of different flow rates: 2, 3, 4, 5, or 6 mL/min and rotation speeds: 600, 800, 1200, and 1600 rpm) [19]. The separation of furanocoumarins presented in CPE between the biphasic systems was monitored by HPLC. In each case, *K* values were calculated. The results are summarized in the Table 1.

Finally, a two-phase solvent system (heptane System VI), composed of *n*Hp:E:M:W (5:2:5:2, *v*/*v*/*v*/*v*), was chosen, where the K for 5-MOP in the ascending mode was close to 3 (K = 2.9636), and it was higher than the K calculated for all of the other methoxyfuranocoumarins. This suggested that in those conditions, 5MOP will be eluted at the end of the CPC chromatogram. In addition, the Kα for each pair of the main furanocoumarins and bergapten was high: 2.6081 (5MOP/8MP) and 4.0404 (5MOP/P), which confirms that bergapten will be separated from the other extract constituents (Table 1). The calculated Kα for the pair of P and 8MP was 1.5493, which enables separation of those two closely related compounds. In the selected conditions (ascending mode), P was predicted to be eluted before 8MP, with both followed by 5MOP. Although, based on the calculated values of the K and Kα coefficients, the descending elution mode in hexane System N° 4 and System N° 5 (Table 1) can be considered suitable for the final isolation and separation of the analyzed coumarins, it nevertheless entails difficulties in the evaporation of the mobile phase (water phase). Moreover, high values of the K coefficient for each of the analyzed compounds resulted in a long elution process time. A similarly long elution time was observed in ascending mode using the heptane System V. Finally, as a result of optimization based on the K values and the Kα coefficients calculated for each pair of analyzed compounds, heptane System VI was selected, in the ascending mode, as being best suited to the effective separation and isolation of the tested methoxyfuranocoumarins from CPE in the hydrostatic CCC process. After optimization of the flow rate and rotation speed, the flow rates of 3 mL/min. and 1600 rpm were finally chosen as the most efficient parameters for separation of 5MOP from the other components in the extract and also for separation of P from 8MP. These parameters were also effective for semi-preparative CPC, where 150 mg of CPE was loaded. The analytical and semi-preparative CPC chromatograms are presented in Figure 2A,B.

### 2.3. HPLC-DAD Purity Assay and ESI-MS Identification of Isolated 5MOP, P, and 8MP

The composition of the CPE and also the purity and identity of the isolated compound were double-checked by HPLC-DAD and by HPLC-DAD-ESI-MS in two different columns and solvent systems. The validated HPLC-DAD method for the quantitative estimation of the purity of the isolated 5MOP, 8MP, and P (*n* = *6*) was applied with use of external standards for targeted compounds. The HPLC analysis of the CPE is presented in the Figure 3.

The calibration curves (in the range 6–220 µg/mL in the six calibration points, *n* = 3) for methoxycoumarins were highly linear ranging from R^2^ = 0.9998 (P and 8MP) to R^2^ = 0.9999 (5MOP). The validation parameters for the methoxyfuranocoumarins are summarized in the Table 2. As was confirmed in the HPLC-DAD analysis after CPC separation, high purity of the isolated compounds was achieved (Table 3). For P: 95.56 and 94.78% (in CPC fractions 11 and 12, respectively); for 8MP: 95.90, 97.24, and 98.14% (in CPC fractions 14, 15, and 16, respectively), and for 5MOP: 95.19 and 100% (in CPC fractions 31 and 32–35, respectively). The identity of the isolated methoxyfuranocoumarins was confirmed additionally by ESI-MS (Table 2). The obtained results were consistent with the literature data [45,49,63,64]. Semipreparative isolation of the 5MOP afforded c.a. 1.20 ± 0.25 mg of 100% pure compound from 150 mg of crude coumarin sediment (CPE) in a single run. The results for the other isolated coumarins were as follows: for peucedanin: 11.80 ± 0.66 mg/150 mg CPE and for 8-methoxypeucedanin: 8.60 ± 0.43 mg/150 mg CPE. The separation procedure was repeated four times, and fractions collected together were tested for the identity and purity of each of the isolated compounds. The final obtained purity (combined after semi-prep CPC) of bergapten was above 99.9% (c.a. 100%); peucedanin was pure as 95.0% and 8-methoxypeucedanin was 97.0% (Table 3).

Figure 4A–C shows the results of the isolation of P, 8MP, and 5Mop by CPC from CPE, monitored by HPLC-DAD (detection of coumarins at 320 nm; detection of possible impurities at 254 and 360 nm).

In the case of each semi-preparative run, the efficiency of isolation of methoxyfuranocoumarins was calculated in % (Table 4). The isolated compounds were subjected to biological studies.

## 3. Discussion

As plant secondary metabolites of plant origin, furanocoumarins are still gaining attention as novel drug candidates, especially for their anti-cancer and chemo-preventive properties and for acting in the area of the central nervous system [33,51,58]. Compounds of natural origin seem to be promising as therapeutics because of fewer unexpected and undesirable interactions with the human body, and relatively low systemic toxicity, are observed as compared to synthetic drugs [65,66,67]. This is one of the reasons why in recent years so much work has focused on the isolation of pure compounds from natural sources, especially plant sources [11,13,58,61]. The isolated compounds of natural origin are then assessed for their pharmacological activity, and in many cases, the observed effects are auspicious, and the work is worthy of the effort [33,49,50,51,68]. However, it is desirable for natural product isolation processes to be more competitive compared to chemical synthesis contaminating the environment with waste products which must next be recycled and disposed of at high cost and risk. Therefore, especially in recent times, scientists are trying to use isolation methods such as CCC which stands out as being environmentally friendly and effective [17,54,58]. The most commonly used types of CCC apparatus are so-called high-speed (HSCCC) or high-performance (HPCCC) models and also centrifugal partition chromatography (CPC) models, where the performance and separation system are not exactly the same due to different separation mechanisms [15,16]. In HSCCC (or HPCCC), this mechanism is based on the variable gravity field produced by a two-axis gyration mechanism (and a rotary seal-free arrangement for the column) and should be recognized as a hydrodynamic system. In CPC, a constant gravity field is produced by a single-axis rotation, together with rotatory seals for supply of the solvent, and separation takes place in cartridges or disks; therefore, CPC is a hydrostatic equilibrium system and can be likened to a static coil [7]. In HSCCC, extraction and settling are therefore time and space separated, opposite to CPC, where these processes are combined inside the cells simultaneously [17], which results in higher operational pressure and enables higher flow rates during the run [19].

Relatively small differences in hydrophobicity between members of one chemical group, as in the case of methoxycoumarins, such as peucedanin and 8-methoxypeucedanin (chemical structures presented in Figure 1), make it difficult to develop a simple and effective method of isolating and purifying them from a complex plant matrix (e.g., *P. tauricum* fruit extracts). The abovementioned compounds differ with only one methoxy group (in the C8 position), which is an electron-donating group. It implies a similar behaviour in the chromatographic system and problems with the co-elution of these compounds.

As we reported previously, an efficient CPC isolation method for separation of similar methoxycoumarins, xanthotoxin and isopimpinellin (which differ also with only one methoxy group), from the crude extract of *Ammi majus* (Apiaceae) fruits was successfully applied [54]. It was achieved after preliminary fractionation of the extract with the LC method, and these methoxyfuranocoumarins were separated from other constituents co-eluted in the fraction LC/IV. As the final result, pure isopimpinellin (100%) and xanthotoxin (98.72%) were obtained in a single ascending mode CPC run (3 mL/min; 1600 rpm, *n*Hx:E:M;W, 10:9:10:8, *v*/*v*/*v*/*v*).

The aim of the presented work was to develop an efficient method for the isolation and separation of the methoxycoumarins P, 8MP, and 5MOP from *Peucedanum tauricum* MB (Apiaceae) fruits by use of carefully optimized centrifugal partition chromatography. To our knowledge, based on the current state of the scientific literature, this is the first report on the use of the CCC hydrostatic mode for the isolation of peucedanin, 8-methoxypeucedanin, and bergapten.

Up to this date, CCC isolation of P and 8MP has been performed from dichloromethane extract from fruits of *Peucedanum luxurians* Tamamsch. in the dynamic HPCCC process of separation, followed by preparative HPLC purification steps [49]. In an experiment, the HPCCC instrument (Dynamic Extractions Ltd., Slough, Berkshire, UK) was equipped with multilayer polytetrafluorethylene coils, one for analytical purposes (22 mL) and the other (136 mL) for semi-preparative purposes. The authors applied a two-phase solvent system: *n*Hx:E:M:W (6:5:6:5, *v*/*v*/*v*/*v*) in the descending mode of elution (3 mL/min, 1600 rpm). As the HPCCC fraction V (eluted in 54–59 min) P was isolated with high purity 99.7%, and this fraction was not subjected to further purification. Fraction IV (minutes 44–46) containing 8MP (purity higher than 80% but lower than 95%) was in the next step subjected to semi-preparative HPLC, and finally 8MP with 99.7% purity was collected.

In the presented work, from the crude extract (CPE), obtained from *Peucedanum tauricum* fruits, the structurally similar methoxyfuranocoumarins P and 8MP were successfully separated by hydrostatic CCC. For this purpose, the biphasic solvent System VI (Table 1) composed of *n*Hp:E:M:W (5:2:5:2; *v*/*v*/*v*/*v*), in the ascending mode of elution (3 mL/min, 1600 rpm), was used, and no additional purification steps were needed. In the single hydrostatic CPC run, these coumarins were gained as pure as 95.6% and 98.1%, respectively. Next, the analytical separation (20 mg of CPE) was easily transferred to the semi-preparative scale (150 mg of CPE) without significant loss of the final quality.

As bergapten (5-methoxypsoralene) is a pharmacologically important plant secondary metabolite, surprisingly, only a few applications concerning the isolation of this compound from plant matrices by CCC have been conducted up until now, and none have been conducted with the use of the hydrostatic CCC method (CPC). Bergapten (5-MOP) was isolated in hydrodynamic CCC (HSCCC) several times with its final purity ranging from 94.7 to 98.3%. For the isolation of this compound, the applied HSCCC biphasic solvent system was used most frequently on the basis of Arizona N containing: *n*-heptane:ethyl acetate:methanol:water (*n*Hp:E:M:W 1:1:1:1, *v*/*v*) [69,70] or with a composition where *n*-heptane was replaced by *n*-hexane (*n*Hx:E:M:W; 1:1:1:1; *v*/*v*/*v*/*v*) [71]. A different composition of the two-phase system was developed for preparative isolation and purification of bergapten from *Cnidium monnieri*, where *n*-heptane was replaced by *n*-hexane, and methanol was replaced by ethanol (*n*Hx:E:Et:W; 5:5:5:5, *v*/*v*/*v*/*v*). In this case, from 500 mg of crude extract of *Cnidium monnieri* in a single run, 45.8 mg of bergapten at 96.5% purity was obtained by use of increasing the flow rate of the mobile phase stepwise from 1 to 2 mL/min after 180 min of elution (final separation time was 480 min) [72].

In some cases, HSCCC separation was followed by preparative HPLC steps to obtain pure compounds. In the new approach, Liu et al., 2014 [59] combined the novel two-dimensional hyphenation of CCC and HPLC (2D CCC × HPLC). This method was used in the isolation of the single 2D separation run of sixteen coumarin derivatives (with purity ranging from 90.1% to 99.5%) from *Peucedanum praeruptorium* Dunn. crude ethanolic extract. By use of this coupled technique, 2.8 mg of 98.3% pure bergapten was obtained. The CCC system composed of *n*-heptane:acetone:water, (31:50:19, *v*/*v*/*v*) was used with a flow rate of 2 mL/min and a revolution speed of 900 rpm (descending mode). The authors successfully applied for the first time the combination of a heart-cutting technique and a stop-and-go protocol in an on-line 2D CCC × HPLC system.

In a similar on-line 2D CCC × HPLC system that was employed with developed a novel fragmentary dilution and turbulent mixing (FD-TM) interface, isolation of coumarins from *Cnidium monnieri* fruit ethanolic extract (500 mg) was carried out. The eight target compounds were isolated (among them, 7.5 mg of bergapten with 98.2% purity). In this case, the HSCCC system was composed of *n*-heptane:acetone:water, (31:50:19, *v*/*v*/*v*), and a flow rate of 2 mL/min and 900 rpm were used (descending mode of elution) [60].

Microwave-assisted extraction (MAE) coupled with CCC and preparative HPLC was applied for isolation of 98.0% pure bergapten (and also other coumarins) from *Angelica pubescentis radix* [61]. Extracts obtained by MAE were fractionated by use of HSCCC (solvent system: *n*Hex:E:M:W (5:5:6.5:3.5; *v*/*v*/*v*; flow rate 2 mL/min, 800 rpm) and the following compounds were isolated by HPLC. Finally, the isolation process took 450 min.

In other cases, two HSCCC separation processes followed one after another. The effective separation of bergapten was achieved from *Cnidium monnieri* (L.) Cusson, where crude coumarins were obtained by ethanol extraction under sonication from dried fruits. The HSCCC with the biphasic solvent systems *n*-hexane-ethyl acetate-ethanol-water (5:5:4:6; *v*/*v*/*v*/*v*—solvent A, and 5:5:6:4; *v*/*v*/*v*/*v*—solvent B) was next successfully performed with stepwise elution: 0–480 min with solvent A and 480–720 min with solvent B. The five coumarins were obtained from 500 mg of the crude extract in a single run, and the purity of the isolated bergapten (45.3 mg) was 94.7% [73]. However, the isolation procedure was in this case time consuming, and elution of all of the coumarins took 13 h.

Walasek et al. [69] fractionated dichloromethane extract from fruits of *Heracleum mantegazzianum* by HPCCC with the use of two solvent systems, Arizona P (6:5:6:5, *v*/*v*/*v*/*v*) and in the next step, Arizona N (1:1:1:1, *v*/*v*/*v*/*v*), (1600 rpm, flow rate 1 mL/min, and 6 mL/min respectively). Finally, several coumarin fractions were obtained. After the first step of isolation, bergapten was found in three fractions (in Fr1 with xanthotoxin, isopimpinellin, and angelicin, in Fr2 with angelicin, and in Fr3 with angelicin and pimpinellin) and was not purified in the second step.

So far no reports on hydrostatic CCC—centrifugal partition chromatography (CPC) isolation of bergapten can be found in the scientific literature. In the presented work, 5MOP was obtained with a high purity of c.a. 100%, which was achieved in the single run, without additional purification steps, which was carried out from the crude coumarin extract (CPE) for the first time. The biphasic solvent system composed of *n*Hp:E:M:W (5:2:5:2; *v*/*v*/*v*/*v*; flow rate 3 mL/min, 1600 rpm; ascending mode) was effective for the separation of this compound from the other methoxyfuranocoumarins presented in the extract. Elongation of the elution time was beneficial and enabled avoidance of additional separation steps and the use of highly specialized prep-HPLC equipment. In addition, the scaling-up of the process from analytical to semi-preparative, with no significant loss in the effectiveness of the separation, were achieved.

For all of the separation steps, the isolation efficiency was calculated. For CPC isolation, as compared with the predicted amount of targeted furanocoumarins in the crude extract injected in each experiment, the CPC isolation efficiency was found to be higher than 79.25%, depending on the isolated compound. It was observed that for the simplest isolated molecule, a with bigger separation factor, the highest efficiency of isolation was obtained. As has previously been underlined, the CPC technique provides minimum absorption loss, almost 100% sample recovery, and high isolation efficiency [6,11,73]. However, as in this experiment, the isolation efficiency was calculated only for fractions–vials with the desired purity of compounds (higher than 95%); therefore, it did not reach 100% in this case, with it being in the range of 79.25–89.55% (for the sum of P, 8MP, and 5MOP, it was c.a. 80%), which was satisfactory as for semi-preparative scale of isolation.

It is worth noticing, that from the *Peucedanum tauricum* plant matrix, the isolation of 5MOP, P, and 8MP by hydrostatic CCC was performed for the very first time. In this study, 8MP was isolated in a single CPC run with a purity of 98.14% and this is the first report concerning the isolation of this methoxyfuranocoumarin in a single hydrostatic CPC run, as well as the isolation of 5MOP with excellent purity. In the presented work, without additional separation steps, finally three methoxyfuranocoumarins were isolated by CPC from crude plant extract with purity as high as 95.56 (for P), 98.14 (for 8MP), and c.a.100% (for 5MOP). CPC was found to be a powerful tool for the isolation of secondary metabolites from the plant matrix with high purity and recovery. Future experiments, also on the preparative scale and with some other plant matrices, will be challenging and are definitely worth undertaking.

## 4. Materials and Methods

### 4.1. Plant Material Source and General Extraction Procedures

Mature fruits of *Peucedanum tauricum* M.B. (Apiaceae, MB-PT/01/2014) were collected in the Botanical Garden of Maria Curie-Skłodowska University (51°16′ N, 22°30′ E, 200 m AMSL, Lublin, Poland) and were identified by plant taxonomist Krystyna Dąbrowska MSc.

Before use, the fruits were dried in a dark place at room temperature (humidity < 30%, temp. c.a. 25 °C). The dried fruits were pulverized and macerated with petroleum ether for 24 h and next, they were extracted exhaustively with the same solvent (for 25 g of plant material, 250 mL of the solvent was used). As the final result, a crude concentrated petroleum ether extract (CPE) was obtained.

For quantitative analysis of furanocoumarins, PLE/ASE extraction was performed [43]. For this purpose, the plant material (4×1 g of fruits) was placed in a stainless steel cell and extracted with methanol in an ASE 100 apparatus (Dionex, Sunnyvale, CA, USA): static cycles, three; cycle duration, 10 min; flush volume, 60%; purge time 100 s; temperature, 110 °C. The obtained extracts were concentrated under vacuum and placed in 20 mL volumetric flasks until analysis. Independently, the crude coumarin sediment obtained by Soxhlet extraction was carefully weighed (4×10 mg) and dissolved in 10 mL of methanol. In each case, before HPLC and MS analysis, the extracts were purified with the use of 0.45 µm Cronus PTFE syringe filters (SMI-LabHut Ltd., Churcham, Gloucester, UK).

### 4.2. Chemicals

Acetonitrile and methanol (HPLC gradient grade) were purchased from J.T. Baker (Deventer, The Netherlands); *n*-hexane (*n*Hx), *n*-heptane (*n*Hp), ethyl acetate (E), and petroleum ether (PE) were purchased from POCH (Gliwice, Poland). An ultrapure water (18.2 MΩ), as obtained from Simplicity (Millipore, Molsheim, France) purification system was used. For the LC-MS experiments, the water and acetonitrile were of LC-MS grade (J.T. Baker). External standards: bergapten (≥98%, Sigma, St. Louis, MO, USA), 8-methoxypeucedanin, and peucedanin (≥99%, isolated in our laboratory by use of LC/prep TLC and assayed by HPLC-DAD-ESI-MS) [43,45] were used in the quantitative analysis.

### 4.3. Isolation, Purification, and Identification of Furanocoumarins

#### 4.3.1. Centrifugal Partition Chromatography

##### CPC Equipment

The CPC instrument employed in the present study was a model Armen SCPC-250-L (Armen Instrument, Saint Ave, France) equipped with an SCPC-250 Teflon column with a total capacity of 250 mL integrated with a gradient flow pump and a UV lamp (Flash 06S DAD 600) operating in various wavelengths; and with a manual injection valve with a 10 mL sample loop. The rotation speed was adjustable from 0 to 3000 rpm. The CPC system was controlled by Armen Glider CPC V5.0a.05 software.

##### Optimization of the CPC Conditions

Optimization of the CPC conditions for the isolation of the targeted compounds (from the other components presented in CPE) by carried out with the use of a shaking tubes test (briefly, 2 mg of CPE was placed into the tube, shaken vigorously, and, after separation of the two phases, the samples from each phase were taken and analysed by HPLC). Next, the partition coefficient (K) values and separation factor (Kα) for each pair of targeted methoxyfuranocoumarins were calculated. Measurement of K was carried out according to the known methodology [62]. Separation factor (K*α*), the ratio of partition coefficients between two solutes, was determined by calculating *K*_1_/*K*_2_ (where *K*_1_ > *K*_2_) in an ascending or descending mode of elution. The idea for the separation of the targeted compound was to find a K value so high (the optimal K value is recommended to be between 0.2 and 3.0) to obtain bergapten (5MOP) eluted alone (separated from the other extract compounds) at the end of the separation process. Simultaneously, the Kα values to separate peucedanin (P) and 8-methoxypeucedanin (8MP) from each other, and from 5MOP at the same time, needed to be as high as possible (the optimal Kα value is recommended to be higher than 1.2, which enables good separation of compounds). Finally, optimization of the flow rate and rotation speed was carried out for the selected separation system and for this purpose, various parameters were tested (600, 800, 1200, and 1600 rpm vs. 2, 3, 4, 5, and 6 mL/min flow rate).

##### CPC Column Equilibration

Each of the tested solvent mixtures were thoroughly equilibrated in a separation funnel at room temperature. Next, the two phases were separated shortly before use. In the applied ascending mode, the aqueous phase was used as the stationary phase, and the organic phase was used as the mobile phase. In each separation, the column was first filled entirely with the stationary phase at a flow rate of 20 mL/min (500 rpm; 20 min.). Then, the upper organic phase was pumped into the column. The sample dissolved in 8 mL of the mixed two phases (4 mL of each) was injected through the injection valve after equilibration of the column. An effluent from the column was continuously monitored by UV detection (at 254 and 320 nm, represented as the upper and the lower line of the CPC chromatogram, respectively), and the peak fractions (32 mL of each in 18 cm-high flasks) were collected by an automatic fraction collector.

#### 4.3.2. HPLC-DAD and ESI-MS Analysis of Furanocoumarins

For the methanol extracts from fruits obtained after ASE extraction, CPE, samples collected in CPC optimization procedure trials, and finally, fractions purified by CPC, all of these were analyzed by HPLC-DAD (the selectivity of the compounds was confirmed by DAD spectra acquired on-line and recorded in the range 190–400 nm; see Figure 3 and Figure 4). The HPLC equipment used was an Agilent 1100 system (with a G1311A QuatPump, a G1315B DAD, a thermostat ALS Therm G1330B, an injection valve Rheodyne with a 20 μL loop, an ALS G1329A autosampler, and a G1322A membrane degasser), equipped with a Zorbax Eclipse XDB C18 column (250 mm × 4.6 mm ID, 5 μm, Agilent Technologies, Santa Clara, CA, USA), and controlled by an Agilent HPLC OpenLab CDS ChemStation. As the mobile phase, methanol (A) in water (B) was used as follows: 0–5 min. 60% A in B; 5–20 min. from 60 to 80% A in B; 20–30 min. 80 to 85% A in B; post time 10 min 60% A in B. The flow rate was 1 mL/min and the temperature was 25 °C. In each case, 10 µL of the sample was injected.

Isolated compounds and corresponding external standards were carefully weighed, and a stock solution of 1 mg of each compound dissolved in 10 mL of methanol was prepared. The working solutions for each calibration curve were prepared in the range 6–220 µg/mL for 5MOP, P, and 8MP and then analyzed by the measurement of their peak areas at λ = 320 nm. The LOD and LOQ were calculated from the calibration plots of the analyzed compounds by applying the formulas: LOD = 3.3σ·a^−1^ and LOQ = 10σ·a^−1^, where (σ) is the standard deviation of the response and (a) is the slope of the calibration curve. The identity of the isolated furanocoumarins was also confirmed in ESI-MS experiments performed with the use of an LC/MS spectrometer (Agilent Technologies, Santa Clara, CA, USA) with MassHunter Software, equipped with an XBridge^TM^ Shield RP18; 100 × 2.1 ID, 3.5 µm (Waters; Milford, MA, USA) column. As the mobile phase (pH 4.5; HCOONH_4_), 1% MeCN (A) and 95% MeCN (B) were used; 1–15 min 30% B; 15–25 min 45% B; 25–35 min 85% B, with a flow rate 0.2 mL/min and the post time took 10 min. The analysis was performed in the mass range of 100–1000 *m*/*z*, Fragmentor 215 V (positive ion mode), capillary voltage 4000 V, nebulizer pressure 35 psi, skimmer 65 V, gas temperature 350 °C, and nitrogen flow 10 L/min. In each case, 2 µL of the sample was injected.

#### 4.3.3. Efficacy of Isolation and Purification Steps

The Soxhlet extraction efficacy of CPE and the isolation efficacy of methoxycoumarins in the single semi-preparative CPC run were calculated (in %). The CPC fractions with the highest purity of targeted compounds were evaporated, dissolved in 10 mL of methanol, and analyzed by HPLC. The obtained results were calculated on 150 mg of CPE (considered to be 100%) loaded in a single semi-preparative run. On this basis, quantitative HPLC results for each methoxyfuranocoumarin were calculated.

In order to determine the efficiency of the applied method, recovery tests were performed. The mixture of pure standards (0.5 mg of each; 5MOP, 8MP, and P) dissolved in 10 mL of MeOH was passed by 0.45 µm PTFE filters and analyzed by HPLC/DAD. The recoveries were carried out by spiking 0.5 mg of each of the tested compounds to the 10 mL of methanolic CPE extract (1 mg of CPE dissolved in MeOH). Next, the samples were passed by the 0.45 µm PTFE filters and analyzed by HPLC/DAD. The analyses were repeated three times.

The recovery data were obtained from the relationship between the amount of standard added and the amount detected and were expressed in %. In the analysis of methanolic ASE, extract recoveries were performed in the same way.

For the tested compounds, the recovery [%] for the pure standards were as follows; 99.7 ± 0.4 for P; 98.9 ± 1.2 for 8MP; and 99.8 ± 0.6 for 5MOP. For the spiked samples, the recoveries were as follows: 98.7 ± 0.4 for P; 97.4 ± 0.8 for 8MP; and 99.7 ± 0.4 for 5MOP in the case of CPE, and 98.9 ± 03 for P; 98.5 ± 0.7 for 8MP; and 99.6 ± 0.5 for 5MOP in the ASE methanolic extract.

In the case of both extracts (ASE and CPE), the recoveries for pure standards were higher than for the spiked samples, highlighting some influence of the matrix; however, the differences were not highly significant.

Statistical analysis was carried out by use of the Student’s *t*-test (*α* = 0.05; *p* = 95%; *n* = 4).

## 5. Conclusions

Rare methoxyfuranocoumarins, peucedanin, 8-methoxypucedanin, and bergapten, were separated by centrifugal partition chromatography (CPC) giving the pure compounds in high yield. The CPC hydrostatic technique was found to be a suitable tool for the isolation and enabled the efficient separation of closely related compounds from the raw plant matrix without additional pre- and/or post-purification steps. The lack of sample loss and the solvent-saving procedure make this CCC mode a convenient and environmentally friendly technique that can be suitable for efficiently isolating high-purity natural products from complex matrices such as, e.g., fruit extracts. Since the analytical CPC separation method can be easily transferred to a semi-preparative scale, a sufficient amount of pharmacologically important plant metabolites can be obtained simply and with good process efficiency. The proposed analytical conditions allow the preparation of these compounds for either industrial or research purposes, such as, for example, pharmacological studies or in vivo tests.

## Figures and Tables

**Figure 1 molecules-28-01923-f001:**
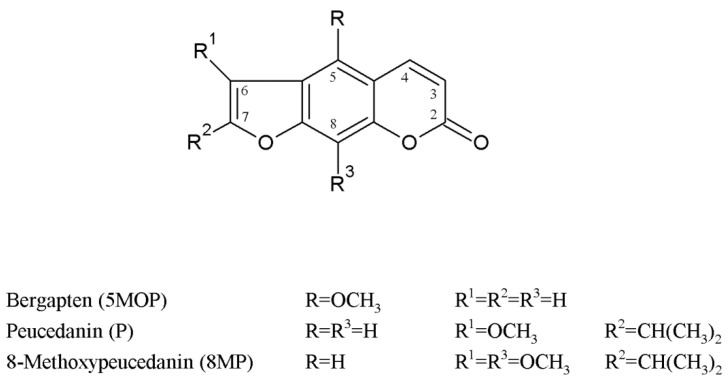
The chemical structure of methoxyfuranocoumarins isolated from *P. tauricum* fruits.

**Figure 2 molecules-28-01923-f002:**
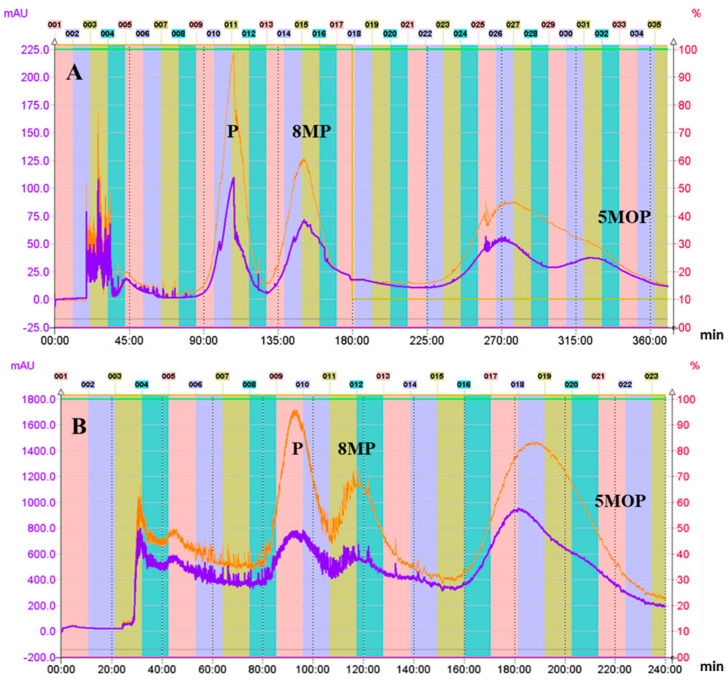
CPC chromatograms of the crude petroleum ether extract of *Peucedanum tauricum* fruits; solvent system: *n*Hp:E:M:W (5:2:5:2, *v*/*v*/*v*/*v*) in the ascending mode of elution (stationary: aqueous phase; mobile: organic phase; flow rate: 3 mL/min; rotation speed: 1600 rpm; detection: 320 nm lower line; 254 nm higher line). Sample size of crude extract: 20 mg on the analytical scale (**A**) and 150 mg on the semi-preparative scale (**B**). The sample was injected without equilibration (**A**) and after equilibration (**B**) of the column. The retention of the stationary phase was c.a. 60–45%.

**Figure 3 molecules-28-01923-f003:**
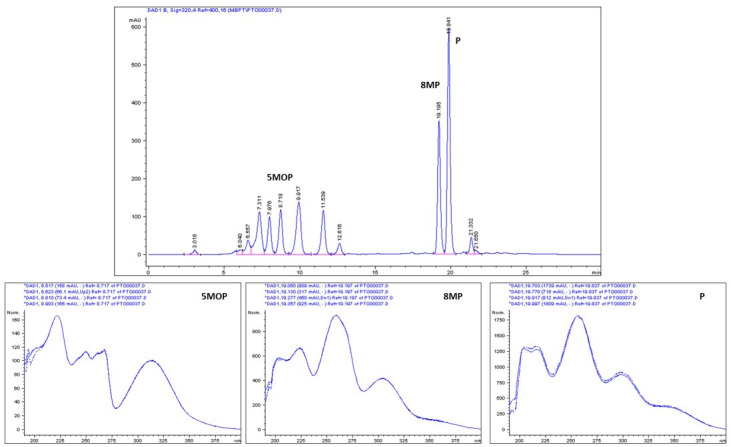
RP-HPLC chromatogram of the crude coumarin extract from *P. tauricum* fruits, with the DAD/UV spectra of targeted methoxyfuranocoumarins: 5MOP (bergapten), 8MP (8-methoxypeucedanin), and P (peucedanin).

**Figure 4 molecules-28-01923-f004:**
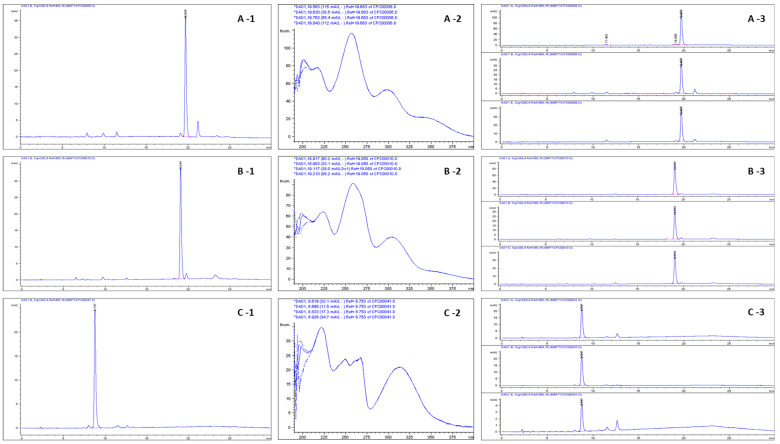
RP-HPLC chromatograms (at 320 nm; (**A-1**,**B-1**,**C-1**)) of methoxyfuranocoumarins isolated by CPC, with their DAD spectra (range 190–400 nm; (**A-2**,**B-2**,**C-2**)). The RP-HPLC chromatograms detected at 254 nm and 360 nm for monitoring of possible impurities are also presented (**A-3**,**B-3**,**C-3**).

**Table 1 molecules-28-01923-t001:** Partition coefficients: *K_asc_* (ascending mode) and *K_desc_* (descending mode) for coumarins in the selected two-phase solvent system and separation factors (*Kα*) for each pair of targeted methoxyfuranocoumarins: bergapten (5MOP), 8-methoxypeucedanin (8MP), and peucedanin (P), presented in crude petroleum ether extract (CPE) of *P. tauricum* fruits.

No	*n*Hp:E:M:W (*v*/*v*/*v*/*v*)	*K_asc_*			*K_desc_*			*K_α_*		
		5MOP	8MP	P	5MOP	8MP	P	*5MOP*/*8MP*	*5MOP*/*P*	*8MP*/*P*
**I**	5:6:5:6	0.5038	0.1593	0.1166	1.9850	6.2773	8.5730	3.1626	4.3188	1.3662
**II**	1:1:1:1	0.4877	0.1564	0.1129	2.0505	6.3951	8.8579	3.1183	4.3198	1.3853
**III**	6:5:6:5	2.4952	0.8678	0.6060	0.4008	1.1524	1.6502	2.8753	4.1175	1.4320
**IV**	3:2:3:2	2.4311	0.9086	0.6193	0.4113	1.1006	1.6148	2.6757	3.9256	1.4671
**V**	2:1:2:1	3.9906	1.5123	0.9990	0.2506	0.6612	1.0010	2.6388	3.9946	1.5138
**VI**	5:2:5:2	**2.9636**	**1.1363**	**0.7335**	0.3374	0.8800	1.3634	**2.6081**	**4.0404**	**1.5493**
**No**	** *n* ** **Hx:E:M:W** **(*v*/*v*/*v*/*v*)**	** *K_asc_* **			** *K_desc_* **			** *K_α_* **		
		**5MOP**	**8MP**	**P**	**5MOP**	**8MP**	**P**	***5MOP*/*8MP***	***5MOP*/*P***	***8MP*/*P***
**1**	10:6:10:6	3.0090	0.7702	0.5710	0.3323	1.2983	1.7513	3.9068	5.2697	1.3489
**2**	10:6:10:7	1.9471	0.5318	0.3981	0.5136	1.8803	2.5118	3.6613	4.8910	1.3359
**3**	10:7:10:6	2.3341	0.6381	0.4843	0.4284	1.5670	2.0650	3.6579	4.8195	1.3176
**4**	10:7:10:7	1.6909	0.5205	0.3300	**0.5914**	**1.9210**	**3.0307**	**3.2486**	**5.1239**	**1.5773**
**5**	10:7:10:8	0.7546	0.1926	0.1285	**1.3252**	**5.1925**	**7.7810**	**3.9180**	**5.8724**	**1.4988**
**6**	10:8:10:7	1.0017	0.9887	0.8131	0.9983	1.0114	1.2298	1.0132	1.2320	1.2160
**7**	10:8:10:8	0.9815	0.9642	0.9718	1.0188	1.0371	1.0289	1.0180	1.0100	1.0079
**8**	10:8:10:9	0.8882	0.2913	0.2401	1.1258	3.4326	4.1641	3.0491	3.6993	1.2131
**9**	10:9:10:8	1.0064	0.3794	0.2961	0.9936	2.6355	3.3767	2.6526	3.3989	1.3038
**10**	10:9:10:9	0.6621	0.2204	0.2010	1.5104	4.5365	4.9748	3.0041	3.2940	1.0965
**11**	*10:10:10:10*	0.5177	0.1475	0.1719	1.9315	6.7813	5.8196	3.5098	3.0116	1.1654

*K_asc_* = *A lph*/*A uph*; *K_desc_* = *A uph*/*A lph*; *A lph*—peak area of the compound in the lower phase; *A uph*—peak area of the compound in the upper phase, *K_α_*—calculated separation factor for each pair of targeted compounds; *K_α_* = *K_II_*/*K_I_*; where *K_II_* > *K_I_*; *n*Hp:E:M:W—*n*-Heptane:Ethyl acetate:Methanol:Water.

**Table 2 molecules-28-01923-t002:** Validation data for quantitative HPLC/DAD analysis and HR-ESI-MS assay for methoxyfuranocoumarins isolated from *P. tauricum* fruits.

Cmp	t_R_ [min] ± SD *n* = *6*	DAD/UV in MeOH [nm]	Calibration Equation y = ax + b (*n* = *3*) a ± SD; b ± SD	Range * 6–220 μg/mL		ESI-MS [H^+^] Fragmentation *m*/*z* (Relative Abundance)
				LOD	LOQ	
5MOP	8.542 ± 0.02 TIC; 9.082	222, 235 sh, 249, 254 sh, 259, 270, 278 sh, 310	y = 34573x + 49.81 a ± 42.13; b ± 4.06 R^2^ = 0.9999	0.45	1.35	C_12_H_8_O_4_, MW 216.0901 ESI-MS: *m*/*z* 217.0974 (100, [M + H]^+^), 189.0254 (10.4), 173.5794 (4.64)
8MP	18.825 ± 0.09 TIC; 20.254	223, 237 sh, 258, 285 sh, 303	y = 61018x + 69.83 a ± 12.08; b ± 1.19 R^2^ = 0.9998	0.84	2.52	C_16_H_16_O_5_, MW 288.0995 ESI-MS: *m*/*z* 289.1068 (100, [M + H]^+^); 259.1153 (61.41), 229.0744 (8.31), 189.0313 (0.19)
P	19.469 ± 0.06 TIC; 20.876	208 sh, 216, 230 sh, 256, 277 sh, 296, 341	y = 46644x + 46.93 a ± 0.18; b ± 7.77 R^2^ = 0.9998	0.79	2.37	C_15_H_14_O_4_ MW, 258.0879 ESI-MS: *m*/*z* 259.0952 (100, [M + H]^+^), 229.0747 (14.94), 189.0314 (0.35)

* Six calibration points: LOD—limit of detection; LOQ—limit of quantification; 5MOP—bergapten; 8MP—8-methoxypeucedanin; P—peucedanin; Cmp—compound.

**Table 3 molecules-28-01923-t003:** Purity [%] (assayed by HPLC-DAD) of methoxyfuranocoumarins, isolated from *P. tauricum* fruits, by analytical and semi-preparative CPC in the biphasic system *n*Hp:E:M:W (5:2:5:2, *v*/*v*/*v*/*v*); in ascending mode (stationary—aqueous phase; mobile—organic phase; rotation speed: 1600 rpm, flow rate: 3 mL/min, sample amount: 20 mg on the analytical scale and 150 mg on the semi-preparative scale). The elution time [min] of isolated compounds collected in CPC fractions (32 mL/vial) is presented.

Analytical Experiment				Semi-Preparative Experiment			
CPC Fraction (vial N^o^) *	Purity [%]			CPC Fraction (Vial N^o^) *	Purity [%]		
	P	8MP	5MOP		P	8MP	5MOP
**11**	95.56			**10**	94.85		
**12**	94.78						
**14**		95.90		**12**		96.97	
**15**		97.24					
**16**		98.14					
**32**			95.19	**21**			98.90
**33**			100	**22**			100
**34**			100	**23**			100
**35**			100				
Time [min]	75–95	100–125	290–335	Time [min]	65–80	85–100	180–210

* Vial volume 32 mL; P—peucedanin; 8MP—8-methoxypeucedanin; 5MOP—bergapten.

**Table 4 molecules-28-01923-t004:** Quantitative determination (*n* = 4) of selected methoxyfuranocoumarins in *P. tauricum* fruits (mg/g dry weight) and in crude petroleum ether extract (mg/g CPE) with the calculation of the isolation efficiency of the single semi-preparative CPC run (in %).

Cmp	mg/g Dry wt. of Plant Material	mg/g of CPE	mg in 150 mg of CPE		CPC Semi-Prep Single Run Isolation Efficiency [%]
			Calculated	Isolated	
5MOP	1.12 ± 0.09	8.94 ± 0.12	1.34 ± 0.45	1.20 ± 0.25	89.55
8MP	8.75 ± 0.37	69.83 ± 0.42	10.48 ± 0.09	8.60 ± 0.43	82.06
P	12.44 ± 0.72	99.27 ± 0.25	14.89 ± 0.76	11.80 ± 0.66	79.25
5MOP + 8MP + P	22.31 ± 0.40	178.04 ± 0.68	26.71 ± 0.43	21.60 ± 0.32	80.87

5MOP—bergapten; 8MP—8-methoxypeucedanin; P—peucedanin; Cmp—compound.

## Data Availability

The data presented in this study are available on request from the corresponding author.

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
