# Peer review of "Efficient Separation of the Methoxyfuranocoumarins Peucedanin, 8-Methoxypeucedanin, and Bergapten by Centrifugal Partition Chromatography (CPC)"

_molecules, 2023, doi:10.3390/molecules28041923_

Round 1

Reviewer 1 Report

The counter-current chromatography in hydrostatic equilibrium system has important significance for separation of natural products. This is a carefully done research paper and the conclusion is of helpful. The theme and results are worth publishing.Minor revisions are recommended.

1. The meaning of high performance liquid chromatography with diode array detection and mass spectrometry is inconsistent with HPLC/DAD/HR-ESI-MS.

2. Carefully examine the writing of the Ca2+ in Page3 line 107. 

3. Figure 3 should be redrawn with professional software.

4. Table 2 typesetting is disorder, it is recommended to modify.

5. Statistical symbols are generally italics, such as n=3, p<0.05.

6.  2D CCC-HPLC system. Or 2D CCC x HPLC system Should be unified ?

7. counter-current chromatography in hydrostatic equilibrium system (CPC) ï¼Ÿ

8. Figure 2 should be added in units of horizontal coordinates.

9. The forms should be standardized.

10. Line 301 for isolation and separation of highly pure ... ?

Author Response

Respected Reviewer,

Thank you kindly for valuable and important insights contained in the review. Below I will try to respond to the comments, hoping for positive reception of my answers and explanations.

Questions and insights;

The counter-current chromatography in hydrostatic equilibrium system has important significance for separation of natural products. This is a carefully done research paper and the conclusion is of helpful. The theme and results are worth publishing. Minor revisions are recommended.

  1. The meaning of high performance liquid chromatography with diode array detection and mass spectrometry is inconsistent with HPLC/DAD/HR-ESI-MS.

Answer 1

I agree with this insight.

In the Abstract and in the main text it was changed and unify as “ high performance liquid chromatography with diode array detection and electrospray ionization mass spectrometry (HPLC-DAD-ESI-MS)” which was exactly the method used in this experiments.

  1. Carefully examine the writing of the Ca2+ in Page3 line 107.

Answer 2

The writing of the Ca2+ was changed, and now is as follows; “”Also activity of this compound as Ca2+ channel blocker was confirmed…”

  1. Figure 3 should be redrawn with professional software.

Answer 3

The Figure 3 was redrawn and I hope that now it is in sufficient quality.

  1. Table 2 typesetting is disorder, it is recommended to modify.

Answer 4

Table was modified to introduce the order.

  1. Statistical symbols are generally italics, such as n=3,p<0.05.

Answer 5

Statistical symbols are now written in italics; in the Table 2; n=6, n=3

and the section 4.3.3.; α=0.05; p=95%; n=4

  1. 2D CCC-HPLC system. Or 2D CCC x HPLC system Should be unified ?

Answer 6

It was unified according to the suggestion and it is now; “2D CCC x HPLC system” as in the reference source addressed to the cited work.

  1. counter-current chromatography in hydrostatic equilibrium system (CPC) ï¼Ÿ

Answer 7

Thank you for this insight – indeed it should be; “counter-current chromatography (CCC) “ or “centrifugal partition chromatography (CPC)”

It was changed in the keywords, and in the text.

  1. Figure 2 should be added in units of horizontal coordinates.

Answer 8

The units - minutes [min] were added to the Figure 2A and B on the each OX axis.

  1. The forms should be standardized.

Answer 9

The forms have been standardized as suggested. Thank you for this observation.

  1. Line 301 for isolation and separation of highly pure ... ?

Answer 10

It was changed as follows; “…for isolation and separation of methoxycoumarins…”

Once again, thank you for detailed comments, which significantly affect the quality of the presentation of my work.

Reviewer 2 Report

1. Consider changing the paper title ;

  "Efficient separation of Methoxyfranocomarins: peucedanin, 8-methoxypeucdanin and bergapten by counter-current chromatography in hydrostatic equilibrium system(CPC)" to  "Efficient separation of Methoxyfranocomarins: peucedanin, 8-methoxypeucdanin and bergapten by centrifugal partition chromatography(CPC)"

2. The authors presented only the results of RP-HPLC analysis of P. tauricm extract (Table 3). However, additional experiments are required for the RP-HPLC analysis results of standard products (5MOP, 8MP, and P) for accurate substance identification.

Author Response

Respected Reviewer,

Thank you kindly for valuable and important insights contained in the review. Below I will try to respond to the comments, hoping for positive reception of my answers and explanations.

  1. Consider changing the paper title ;

  "Efficient separation of Methoxyfranocomarins: peucedanin, 8-methoxypeucdanin and bergapten by counter-current chromatography in hydrostatic equilibrium system(CPC)" to  "Efficient separation of Methoxyfranocomarins: peucedanin, 8-methoxypeucdanin and bergapten by centrifugal partition chromatography(CPC)"

Answer 1

The title of the manuscript was changed as suggested by Reviewer;

 "Efficient separation of Methoxyfuranocoumarins: peucedanin, 8-methoxypeucedanin and bergapten by centrifugal partition chromatography (CPC)"

  1. The authors presented only the results of RP-HPLC analysis of P. tauricum extract (Table 3). However, additional experiments are required for the RP-HPLC analysis results of standard products (5MOP, 8MP, and P) for accurate substance identification.

Answer 2

The results of HPLC/DAD analysis of 5MOP, P and 8MP are available because they have been done during experiments, however they were not presented in the Manuscript because of lot of data presented already. HPLC chromatograms of each isolated compound with DAD spectra (range 190-400 nm) collected on-line in the time of analysis from photodiode array detector are now included and presented as the Figure 4 A-C. It should be also mentioned that during analysis spectra were recorded for coumarins at 320 nm and also at 254, 360 nm for possible interfering compounds, detecting none of them on chromatograms of isolated furanocoumarins. This information was also placed in the text in section Results. 

Reviewer 3 Report

1. what's the final target of separation, peucedanin, 8-methoxypucedanin and bergapten?

2. what is the importance of the methoxyfuranocoumarins, it would be more clear in the introduction

3. did the CPC has the broad separtion ability to any other chemical compounds, is that any discussion in the study?

Author Response

Respected Reviewer,

Thank you kindly for valuable and important insights contained in the review. Below I will try to respond to the comments, hoping for positive reception of my answers and explanations.

Question 1. what's the final target of separation, peucedanin, 8-methoxypucedanin and bergapten?

Answer 1

Furanocoumarins and among them methoxyfuranocoumarins as compounds from natural sources gained attention as active metabolites, therapy adjuvants, basis for developing new drugs. Recently e.g. some antimicrobial derivatives of peucedanin (isolated from Peucedanum morissoni) were developed and tested as antimicrobial agents (Molecules 2019, 24, 2126; doi:10.3390/molecules24112126).

Other coumarins, such as e.g. xanthotoxin (8-methoxypsoralen) has influence on cancer cell lines and induces intrinsic and extrinsic apoptotic pathways, suppresses cell growth of SK-N-AS neuroblastoma and SW620 metastatic colon cancer cells ([in the text of the Manuscript it was ref 33 ]J. Ethnopharmacology, 2017; 207; 19-29; https://doi.org/10.1016/j.jep.2017.06.010).

This is the reason why separation and isolation of single methoxyfuranocoumarins is important, and a final target of isolation performed in this work is to use isolated compounds in biological studies, especially in vitro tests on e.g. over mentioned cancer cell lines.

Question 2. what is the importance of the methoxyfuranocoumarins, it would be more clear in the introduction

Answer 2

The fragment underlying and clarifying the importance of the methoxyfuranocoumarins was added (with the supporting references) in the Introduction section, as was recommended.

“Some methoxycoumarins are important plant drugs, such as e.g xanthotoxin (8-methoxypsoralen), used in psoriasis and vitiligo treatment since many years. Recently influence of this natural substance on apoptosis induction by intrinsic and extrinsic pathways and suppression of cell growth of SK-N-AS neuroblastoma and SW620 metastatic colon cancer cells were found.”

Question 3. did the CPC has the broad separtion ability to any other chemical compounds, is that any discussion in the study?

Answer 3

As the CPC is an efficient tool for separation of many groups of natural products (some reviews are available as listed below, and some are cited in the Introduction), there is no broad discussion about it in the Discussion section, focusing only for isolation of peucedanin, 8-methoxypeucedanin and bergapten by HSCCC and CPC. However among coumarins some compounds were successfully isolated by CPC e.g. xanthotoxin and isopimpinellin (ref. 54 in the manuscript). Other natural compounds as e.g. curcuminoids (ref 17 in the manuscript), flavonoids and essential oil constituents were also efficiently isolated by CPC. The examples of hyphenation of CPC with HSCCC were also published recently, highlighting isolation of alkylresorcinols.

Review examples;

Application of CPC and related methods for the isolation of natural substances - a review. By: Kedzierski, B.; Kukula-Koch, W.; Glowniak, K. ; Acta Poloniae Pharmaceutica (2014), 71(2), 223-227.

Techniques for extraction and isolation of natural products: a comprehensive review. By: Zhang, Qing-Wen; Lin, Li-Gen; Ye, Wen-Cai; Chinese Medicine (London, United Kingdom) (2018), 13, 20/1-20/26 - (page 14 – about separation of volatile constituents by CPC).

Countercurrent Separation of Natural Products By: Pauli, Guido F.; Pro, Samuel M.; Friesen, J. Brent; Journal of Natural Products (2008), 71(8), 1489-1508 (reference 13 in reviewed article)

Isolation of polyphenols;

Development of "ultrasound-assisted dynamic extraction" and its combination with CCC and CPC for simultaneous extraction and isolation of phytochemicals

By: Zhang, Yuchi; Liu, Chunming; Li, Jing; Qi, Yanjuan; Li, Yuchun; Li, Sainan ; Ultrasonics Sonochemistry (2015), 26, 111-118

 Bryophyllum pinnatum markers: CPC isolation, simultaneous quantification by a validated UPLC-DAD method and biological evaluations; By: Morais Fernandes, J. et al.;

Journal of Pharmaceutical and Biomedical Analysis (2021), 193, 113682

 Examples of hyphenation of CPC with HSCCC published recently;

Online hyphenation of centrifugal partition chromatography with countercurrent chromatography (CPC-CCC) and its application to the separation of saturated alkylresorcinols By: Hammerschick, Tim; Vetter, Walter; Analytical and Bioanalytical Chemistry (2022), 414(17), 5043-5051

Yours faithfully

Magdalena Bartnik

Reviewer 4 Report

The paper is put in a very good context and I applaud the author for the thorough introduction. It is generally well written but would benefit from some proofreading and format corrections.

Spacing in sec 2.2 differs from that in 2.1

There are quite a few unidentified peaks if Figure 3, could the author comment on what they might be?

The following is the major issue with the manuscript:

The author refers to a good recovery %, how what the original amount in the crude extract determined? No spiking procedures were performed. I would suggest adding known amounts of standards of the compounds of interest (or isotopic labeled compounds or different compounds with similar properties). With the current data, it is unclear to me how the author can say anything about the true recovery rate.

The standard curves for the data in Table 2 could also be misleading. According to the materials/methods the authentic standards were dissolved in pure methanol which would not give the same type of matrix interference as for the isolated compounds. As shown in the previous data, many unknown species co-eluted with the compounds of interest and may contribute to and/or mask signals attributed to the analytes of interest. This is why a recovery/spiking experiment is essential to gauge the level of matrix effect on the measurements.

Author Response

Respected Reviewer,

Thank you kindly for valuable and important insights contained in the review. Below I will try to respond to the comments, hoping for positive reception of my answers and explanations.

The paper is put in a very good context and I applaud the author for the thorough introduction. It is generally well written but would benefit from some proofreading and format corrections.

  1. Spacing in sec 2.2 differs from that in 2.1

Answer 1

Thank you for this observation. Indeed it is, but I tried to change it a few times and it was hard to change, maybe because of some problems with Templete. This could also be my mistake. The editing, at the end (hopefully it will be also the final editing for my Manuscript) will resolve some technical issues as the manuscript is, at the end, edited by Professionals.

  1. There are quite a few unidentified peaks if Figure 3, could the author comment on what they might be?

The HPLC chromatogram on the Figure 3 presents coumarin fraction in the CPE. As the extract was obtained by petroleum ether during the Soxhlet extraction mostly the lipophilic compounds are present. The extract obtained from P. tauricum mature fruit was in detail analysed previously (among other extracts from different aboveground parts of this plant; from immature fruits, leaves and flowers). A few coumarins were identified such as; scopoletin (mainly in leaves), oxypeucedanin hydrate (traces in mature fruit), oxypeucedanin (very low amount c.a.0.28 mg/g dr wt in mature fruit), and bergapten, 8-methoxypeucedanin, peucedanin and isoimperatorin. The main components of the extract from mature fruit are peucedanin (c.a. 13.5 mg/g dr wt) and 8-methoxypeucedanin (c.a. 6.5 mg/g dr wt, followed by bergapten (c.a. 1.0 mg/g dr wt) and isoimperatorin (0.14 mg/g dr wt).

In mature fruit c.a. 0.11 mg/g dr wt of phenolic acids were found (absent in the petroleum ether fraction, as they are hydrophilic compounds), 1.39 mg/g dr wt of flavonoids (also not presented in the petroleum ether fraction).

The possible presence of interfering compounds in the extract was monitored by DAD detection at 245, 360 nm simultaneously with the detection of coumarins (at 320 nm).

Also for each isolated coumarin the detection was done this way, to find out if impurities are presented together with the isolated compounds. In revised version of the manuscript Figure 4 A-C was added presenting DAD spectra and HPLC analysis of the single isolated compounds.

  1. The following is the major issue with the manuscript:

The author refers to a good recovery %, how what the original amount in the crude extract determined? No spiking procedures were performed. I would suggest adding known amounts of standards of the compounds of interest (or isotopic labeled compounds or different compounds with similar properties). With the current data, it is unclear to me how the author can say anything about the true recovery rate.

The standard curves for the data in Table 2 could also be misleading. According to the materials/methods the authentic standards were dissolved in pure methanol which would not give the same type of matrix interference as for the isolated compounds. As shown in the previous data, many unknown species co-eluted with the compounds of interest and may contribute to and/or mask signals attributed to the analytes of interest. This is why a recovery/spiking experiment is essential to gauge the level of matrix effect on the measurements.

Answer3

I couldn’t agree more. When quantitative analysis of new matrix (or new method) is performed it is necessary to perform recovery studies by spiking added substance to the matrix (extract) and comparing it with pure standards, to find out how the matrix affected the isolation of compounds. It is the way of measuring the efficacy of the method. Thank you very much for underlying this aspect of validation, and highlighting, that presentation of this data shouldn’t be omitted.

The recovery tests on Peucedanum tauricum plant matrix were previously done (see ref. cited in the manuscript;  Bartnik, M.; GÅ‚owniak, K. Furanocoumarins from Peucedanum tauricum Bieb. and their variability in the aerial parts of the plant during development. Acta Chromatogr. 2007, 18, 5-14). It was done in to the accordance to the method described previously by Authors working on the coumarins containing matrices (see reference; Phytochem. Anal. 10, 268–271, (1999) by Zgórka G and GÅ‚owniak K.).

Therefore also in the present study the analysis of the ASE extract and also CPE for recovery of 5MOP, 8MP and P were completed.

The reason why it was not inserted in the v_1 of the Manuscript was the slightly low difference for standards and for spiked samples counted in the experiments, and also multiple data inserted already in the described work. Now I have placed this data in the text in the section 4.3.3.

The additional information are as follows;

“In order to determine the efficiency of the applied method the recovery tests were performed. The mixture of pure standards (0.5 mg of each; 5MOP, 8MP and P) dissolved in 10 mL of MeOH was passed by the 0.45 µm PTFE filters and analysed by HPLC/DAD. The recovery were done by spiking 0.5 mg of the each tested compound to the 10 mL of methanolic CPE extract (1 mg of CPE dissolved in MeOH). Next samples were passed by the 0.45 µm PTFE filters and analysed by HPLC/DAD. The analyses were repeated 3 times.

The recovery data were obtained from the relationship between the amount of standard added and the amount detected and were expressed in %. In the analysis of methanolic ASE extract recoveries were performed in the same way.

For the tested compounds the recovery [%] for pure standards were as follows; 99,7 ± 04 for P; 98,9 ± 1,2  for 8MP and 99,8 ±0.6  for 5MOP. For spiked samples recoveries were as follows; 98.7 ± 0.4  for P, 97.4 ± 0.8 for 8MP, and 99.7 ± 0.4 for 5MOP in case of CPE, and 98,9 ± 03 for P; 98,5 ± 0.7  for 8MP and 99,6 ±0.5  for 5MOP in ASE methanolic extract.

In case of both extracts (ASE and CPE) recoveries for pure standards were higher than for spiked samples, highlighting some influence of the matrix, however, the differences were not highly significant.”

Yours faithfully

Magdalena Bartnik

Round 2

Reviewer 4 Report

Thank you for the response and inclusion of the new section.